# A Data-Efficient Visual-Audio Representation with Intuitive Fine-tuning for Voice-Controlled Robots

**Peixin Chang, Shuijing Liu, Tianchen Ji, Neeloy Chakraborty,**
**Kaiwen Hong, Katherine Driggs-Campbell**
University of Illinois at Urbana-Champaign
{pchang17, sliu105, tj12, neeloyc2, kaiwen2, krdc}@illinois.edu

**Abstract:** A command-following robot that serves people in everyday life must continually improve itself in deployment domains with minimal help from its end users, instead of engineers. Previous methods are either difficult to continuously improve after the deployment or require a large number of new labels during fine-tuning. Motivated by (self-)supervised contrastive learning, we propose a novel representation that generates an intrinsic reward function for command-following robot tasks by associating images with sound commands. After the robot is deployed in a new domain, the representation can be updated intuitively and data-efficiently by non-experts without any hand-crafted reward functions. We demonstrate our approach on various sound types and robotic tasks, including navigation and manipulation with raw sensor inputs. In simulated and real-world experiments, we show that our system can continually self-improve in previously unseen scenarios given fewer new labeled data, while still achieving better performance over previous methods.

**Keywords:** Command Following, Multimodal Representation, Reinforcement Learning, Human-in-the-Loop

## 1 Introduction

Audio command following robots is an important application that paves the way for non-experts to intuitively communicate and collaborate with robots in their daily lives. Ideally, a command-following robot should ground both speech and non-speech commands to visual observations and motor skills. For example, a household robot must open the door when it hears a doorbell or someone saying "open the door." The robot should also be customizable and continually improve its interpretation of language and skills from non-experts [1, 2].

The need for command-following robots has spurred a wealth of research. Learning-based language grounding agents were proposed to perform tasks according to visual observations and text/speech instructions [3, 4, 5, 6, 7]. However, these approaches often fail to completely solve a common problem in learning-based methods: performance degradation in a novel target domain, such as the real world [8, 9, 10]. Fine-tuning the models in the real world is often expensive due to the requirement of expertise, extra equipment and large amounts of labels, none of which can be easily provided by non-expert users in everyday environments. *Without enough domain expertise or abundant labeled data, how can we allow users to customize such robots to their domains with minimal supervision?* Some prior works have attempted to reduce data usage when fine-tuning the robot in new domains. However, the efficiency of the methods usually relies on task-specific assumptions [11], extra sensor instrumentation [12], and limited task variations [13, 14].

In this paper, we propose a novel framework that builds on (self-)supervised contrastive learning to realize more *effective* training and more *efficient* fine-tuning for rewards and skills learning. As shown in Fig. 1, we first learn a joint **V**isual and **A**udio **R**epresentation, which is **D**ata-efficient and can be **I**ntuitively **F**ine-tuned(Dif-VAR). In the second stage, we use the representation to compute

7th Conference on Robot Learning (CoRL 2023), Atlanta, USA.

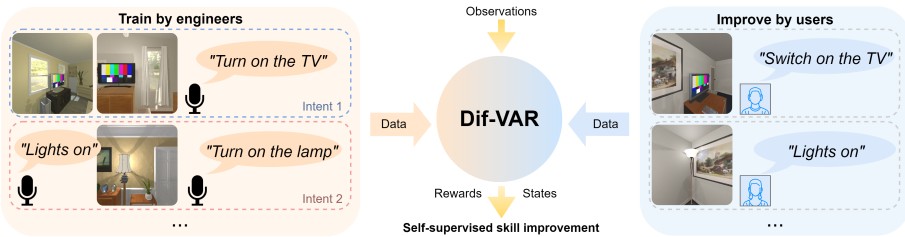

Figure 1: **Our pipeline.** Contrastive learning is used to group images and audio commands of the same intent. The resulting Dif-VAR supports the downstream RL training by encoding the auditory and visual signals and providing reward signals and states to the agent. Users improve the Dif-VAR intuitively by providing the visual-audio pairs in their own domain, and the updated Dif-VAR supervises the RL fine-tuning.

intrinsic reward functions to learn various robot skills with reinforcement learning (RL) without any reward engineering. When the robot is deployed in a new domain, the fine-tuning stage is data efficient in terms of label usage and is natural to non-experts, since users only need to provide a relatively small number of images and their corresponding sounds. For example, a user can teach Dif-VAR by saying that "this is an apple" when the robot sees an apple. Then, RL policies are self-improved with the updated Dif-VAR.

We apply this learning approach to diverse navigation and manipulation environments. Given a sound command, the robot must identify the commander's goal (intent), draw the correspondence between the raw visual and audio inputs, and develop a policy to finish the task. To develop a generally applicable pipeline, we make minimal assumptions. We do not assume the availability of human demonstrations, prior knowledge about the environment, or image and sound recognition modules that have perfect accuracy after the deployment. The robot is equipped with only a monocular uncalibrated RGB camera and a microphone. Our method works with various sound signals (e.g. voice, environmental sound) and various types of robots (e.g. mobile robots, manipulators).

Our main contributions are: (1) We propose a voice-controlled robot pipeline for everyday household tasks. This framework is intuitive for non-experts to continually improve and is agnostic to a wide range of voice-controlled tasks. (2) Inspired by (self-)supervised contrastive loss, we propose a novel representation of visual-audio observations named Dif-VAR, which generates intrinsic RL rewards for robot skill learning. Only image-audio pairs are required to fine-tune Dif-VAR and the RL policy. No reward engineering, state estimation, or other supervision is needed. (3) We propose a variety of voice-controlled navigation and manipulation benchmarks. In new simulated and real-world domains, the adaptation of our method outperforms state-of-the-art baselines in terms of performance and data efficiency, while requiring less expertise and environmental instrumentation.

## 2 Related Works

**End-to-end language understanding.** End-to-end spoken language understanding (SLU) systems extract the speaker's intent directly from raw speech signals without translating the speech to text [15, 16, 17]. Such an end-to-end system is able to fully exploit subtle information, such as speaker emotions that are lost during speech-to-text transcription, and outperform pipelines that preprocess the speech into text [16, 17]. However, they have not been widely applied in robotics.

**Command following agents.** Conventional command following agents consist of independent modules for language understanding, language grounding, and planning [18, 19, 20]. But these modular pipelines suffer from intermediate errors and do not generalize beyond their programmed domains [21, 22, 23]. To address these problems, end-to-end command following agents are used to perform tasks according to text-based natural language instructions and visual observations [3, 4, 23, 24, 25]. In addition, large language models are applied to program the robots given the language prompts [26, 27]. However, these methods neglect the non-speech commands and abstract away the practical challenges of auditory signal grounding. To make full use of audio commands, Chang et al. [7] introduces an RL framework for skill learning directly from raw sounds.

However, all these works overlook the continual fine-tuning and customization by users, an essential step to ensure performance after the deployment. Fine-tuning such models is computationally challenging and requires hand-tuned reward functions and prohibitive labeling efforts. Chang et al. [1] partially addresses the problem by learning a visual-audio representation (VAR) with triplet loss to generate an intrinsic reward function for RL. However, this method requires negative pairs in the triplet loss, which is not intuitive to non-experts and inefficient to deploy in new target domains.

**Visual and language representation for robotics.** Representation learning has shown great potential in learning useful embeddings for downstream robotic tasks [28, 29]. Deep autoencoders have been used to learn a latent space which generates states or rewards for RL [30, 31, 32]. Contrastive learning has also been used to learn representations for downstream skill learning [33, 34]. However, in task execution, a goal image has to be provided, which is less natural in terms of human-robot communication compared to language.

To this end, visual-language representations have been widely used to associate human instructions with visual observations in navigation [27, 35] and manipulation [12, 36, 37]. However, similar to text-based command following agents, these methods also lose information when the input is sound instead of text. Although audio-visual representations such as AudioCLIP have been developed [38], how to apply them in robot learning remains an open challenge. Our work and [1] address this challenge by proposing a visual-audio representation that generates RL rewards for robot skill learning, while our method achieves better data efficiency and can be more easily fine-tuned than [1].

# 3 Methodology

In this section, we describe the two-stage training pipeline and fine-tuning procedure. In training, we assume the availability of sufficiently large labeled datasets, simulators, and labels. However, in fine-tuning, speech transcriptions, one-hot labels, and reward functions, are not available.

## 3.1 Visual-audio representation learning

In the first stage, we collect visual-audio pairs from the environment. Then, we learn a joint representation of images and audios that associates an image with its corresponding sound command.

**Data collection.** Suppose there are $M$ possible intents or tasks within an environment. We collect visual-audio pairs defined as $(\mathbf{I}, \mathbf{S}, y)$ from the environment, where $\mathbf{I} \in \mathbb{R}^{n \times n}$ is an RGB image from the robot's camera, $\mathbf{S} \in \mathbb{R}^{l \times m}$ is the Mel Frequency Cepstral Coefficients (MFCC) [39] of the sound command, and $y \in \{0, 1, ..., M\}$ is the intent ID. We call $\mathbf{I}$ and $\mathbf{S}$ two *views* of an intent $y$. A visual-audio pair contains a goal image and a sound command of the same intent. For example, when an iTHOR agent sees a lit lamp, the agent hears the sound "Switch on the lamp" from the environment. In contrast, when the agent does not see any object in interest or is far away from all objects so that it sees multiple objects at once, it receives only an image and hears no sound. The image is paired with $\mathbf{S} = \mathbf{0}_{l \times m}$ and $y = M$. We define this situation as an *empty intent*.

**Training Dif-VAR.** Our goal is to encode both visual and auditory signals into a joint latent space, where the embeddings from the same intents are pulled closer together than embeddings from different intents. For example, the embedding of an image with a TV turned on needs to be close to the embedding of a sound command "Turn on the TV" but far away from other irrelevant commands such as "Turn off the light." We adopt the idea from (self-)supervised contrastive learning for visual representations and formulate the problem as metric learning. As shown in Fig. 2a, the Dif-VAR is a double-branch network with two main components. The first component contains the encoders $f^I : \mathbb{R}^{n \times n} \to \mathbb{R}^{d_I}$ and $f^S : \mathbb{R}^{l \times m} \to \mathbb{R}^{d_S}$ which map an input image $\mathbf{I}$ and a sound signal $\mathbf{S}$ to representation vectors $\mathbf{h}^I$ and $\mathbf{h}^S$, respectively. In practice, any deep models for image and sound processing can be used for $f^I$ and $f^S$. The second component is the projection heads $g^I : \mathbb{R}^{d_I} \to \mathbb{R}^d$, $g^S : \mathbb{R}^{d_S} \to \mathbb{R}^d$, and $b^I : \mathbb{R}^{d_I} \to \mathbb{R}$ that map the representations $\mathbf{h}^I$ and $\mathbf{h}^S$ to the space where losses are applied. We denote the vector embeddings $g^I(\mathbf{h}^I)$ and $g^S(\mathbf{h}^S)$ as $\mathbf{z}^I$ and $\mathbf{z}^S$,

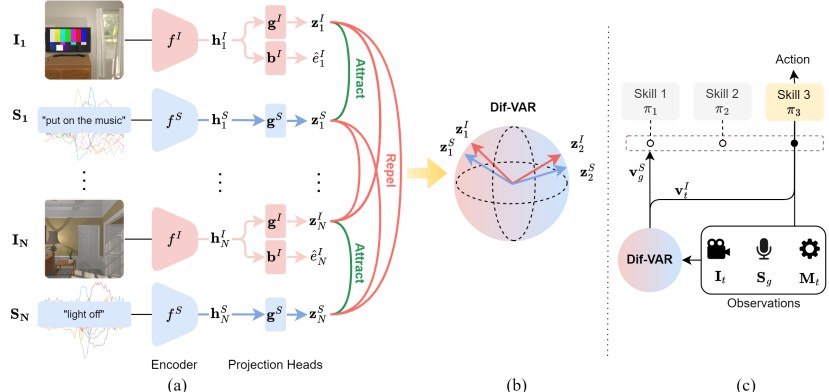

Figure 2: **Purposed framework**. (a) The Dif-VAR is a double-branch network optimized with (self-)supervised contrastive loss. (b) The latent space of the Dif-VAR is a unit hypersphere such that the images and audios of the same intent are closer than those of different intent in the space. (c) The Dif-VAR decides which skill to activate according to $\mathbf{S}_g$.

respectively. We enforce the norm of $\mathbf{z}^I$ and $\mathbf{z}^S$ to be 1 by applying an $L2$-normalization, such that the embeddings live on a unit hypersphere as shown in Fig. 2b.

We use supervised contrastive (SupCon) loss as the objective, which encourages the distance between $\mathbf{z}^I$ and $\mathbf{z}^S$ of the same intent to be closer than those of a different intent [40]. Suppose there are $N$ visual-audio pairs in a batch. Let $k \in K := \{1, ..., 2N\}$ be the index of an image or a sound signal within that batch and $P(k) := \{p \in K \setminus \{k\} : y_p = y_k\}$ be the set of indices of all images and sounds of the same intent except for index $k$. Then, the SupCon loss is

$$\mathcal{L}_{\text{SupCon}} = -\sum_{k \in K} \frac{1}{|P(k)|} \sum_{p \in P(k)} \log \frac{\exp\left(\mathbf{z}_k \cdot \mathbf{z}_p / \tau\right)}{\sum_{j \in K \setminus \{k\}} \exp\left(\mathbf{z}_k \cdot \mathbf{z}_j / \tau\right)}, \tag{1}$$

where $|\cdot|$ is the cardinality, $\mathbf{z}^*$ can be either $\mathbf{z}^I$ or $\mathbf{z}^S$, and $\tau \in \mathbb{R}^+$ is a scalar temperature parameter. The use of SupCon loss allows attraction and repulsion among all images and sound within a batch, which improves the training efficiency of the representation. We introduce a binary classification loss for the image to distinguish between empty and non-empty intent. Let $\mathcal{L}_{\text{BCE}}$ denote the binary cross entropy loss and $e$ denote the label of intent, which is 0 for empty intent and 1 for non-empty intent. The batch loss for training the Dif-VAR is:

$$\mathcal{L}_{\text{Dif-VAR}} = \alpha_1 \mathcal{L}_{\text{SupCon}} + \alpha_2 \frac{1}{N} \sum_{j=1}^{N} \mathcal{L}_{\text{BCE}}(b^I(\mathbf{h}_j^I), e_j) \tag{2}$$

where $\alpha_1$ and $\alpha_2$ are the weights of losses. Depending on if the intent is predicted empty or not, the output $\mathbf{v}^I$ and $\mathbf{v}^S$ of Dif-VAR can be determined for image and sound by:

$$\mathbf{v}^I = \mathbb{1}_{\{\hat{e}^I \geq 0.5\}} \mathbf{z}^I, \quad \hat{e}^I := b^I(\mathbf{h}^I), \quad \mathbf{v}^S = \mathbb{1}_{\{\mathbf{S}_i \neq \mathbf{0}_{l \times m}\}} \mathbf{z}^S. \tag{3}$$

where $\mathbb{1}$ is an indicator function. The purpose of the binary classification is to set the image and sound embeddings of the empty intent to the center of the joint latent space. This centralization removes the biases caused by the location of the empty intent in the joint latent space, leading to better intrinsic reward then previous methods.

While SupCon loss and other self-supervised visual representation learning frameworks are originally only applied to image modality [40, 41], we extend the framework to a multi-modality setting and create a new representation for command following robots.

## 3.2 RL with visual-audio representation

The second stage of our pipeline is to train an RL agent using an intrinsic reward function generated by a trained Dif-VAR. We model a robot command following task as a Markov Decision Process (MDP), defined by the tuple $\langle \mathcal{X}, \mathcal{A}, P, R, \gamma \rangle$. At each time step $t$, the agent receives an image

$\mathbf{I}_t$ from its RGB camera, and robot states $\mathbf{M}_t$ such as end-effector location or previous action. At $t = 0$, an additional one-time sound command $\mathbf{S}_g$ containing an intent is given to the robot. We freeze the trainable weights of Dif-VAR in this stage and define the MDP state $x_t \in \mathcal{X}$ as $x_t = [\mathbf{I}_t, \mathbf{v}_t^I, \mathbf{v}_g^S, \mathbf{M}_t]$, where $\mathbf{v}_t^I$ and $\mathbf{v}_g^S$ are the output of the Dif-VAR for $\mathbf{I}_t$ and $\mathbf{S}_g$, respectively. Then, based on its policy $\pi(a_t|x_t)$, the agent takes an action $a_t \in \mathcal{A}$. In return, the agent receives a reward $r_t \in R$ and transitions to the next state $x_{t+1}$ according to an unknown state transition $P(\cdot|x_t, a_t)$. The process continues until $t$ exceeds the maximum episode length $T$, and the next episode starts.

**Intrinsic rewards.** Since $\mathbf{v}^I$ and $\mathbf{v}^S$ of the same intent are pulled together within the Dif-VAR by the contrastive loss, intrinsic rewards can be derived as the similarity between $\mathbf{v}^I$ and $\mathbf{v}^S$. Eq. 4 and 5 present two possible task-agnostic and robot-agnostic reward functions:

$$r_t^i = \mathbf{v}_t^I \cdot \mathbf{v}_g^S \quad (4) \qquad r_t^{ic} = \mathbf{v}_t^I \cdot \mathbf{v}_g^S + \mathbf{v}_t^S \cdot \mathbf{v}_g^S \quad (5)$$

where $\mathbf{v}_t^S$ is the embedding of the current sound signal $\mathbf{S}_t$, which can be triggered in the same way as $\mathbf{S}$ as described in Section 3.1. Intuitively, the agent using $r_t^i$ receives high reward when the scene it sees matches the command it hears. The agent trained using the reward $r_t^{ic}$ additionally needs to match the current sound it hears with the sound command to receive high rewards. Compared to $r_t^{ic}$, the reward function $r_t^i$ does not depend on any real-time supervision signal such as current sound $\mathbf{v}_t^S$ from the environment, allowing the agent to perform *self-supervised* RL training with Dif-VAR. Although RL agents trained with Eq. 4 can already achieve decent performance, providing the current sound $\mathbf{S}_t$ can further improve the performance [1]. Since $\mathbf{S}_t$ can be difficult to obtain especially in real environments, $\mathbf{S}_t$ is not part of the state $x_t$ and thus the robot policy does not require $\mathbf{S}_t$ at test time.

**Policy network.** We purpose two architectures for RL policies. The first architecture is flat and uses a single policy network to fulfill all the intents in an environment, which is suitable when the skills to finish tasks are similar. The second architecture is hierarchical and contains multiple policy networks individually designed to fulfill a subset of intents in an environment, as shown in Fig. 2c. Given an $\mathbf{S}_g$ and a set of policies $\Pi = \{\pi_1, \pi_2, ...\}$, the Dif-VAR selects a policy $\pi_j$ by

$$j = L(\hat{y}), \quad \hat{y} = \arg\max_i \frac{\mathbf{v}_g^S \cdot C_i}{\|C_i\|_2}. \tag{6}$$

where $C_i$ is the centroid of an intent in the joint latent space calculated from the training data and $L$ is a lookup table that maps an intent ID to a policy. For benchmarking purposes, we use Proximal Policy Optimization (PPO) for policy and value function learning [42].

### 3.3 Intuitive and data-efficient fine-tuning

After the robot is deployed in a new domain such as the real world, its performance often degrades due to domain shift from both perception and dynamics [43]. Our fine-tuning procedure allows non-experts to *continually* improve the Dif-VAR to reduce perception gaps and improve robot skills to reduce dynamics gaps. We only need to collect visual-audio pairs of the form $(\mathbf{I}, \mathbf{S})$ from non-experts. Since we no longer have the underlying labels $y$ for images and sounds, we replace the SupCon loss in Eq. (2) with the following self-supervised contrastive loss (SSC) [41]:

$$\mathcal{L}_{\text{SSC}} = -\sum_{k \in K} \log \frac{\exp\left(\mathbf{z}_k \cdot \mathbf{z}_{p(k)}/\tau\right)}{\sum_{j \in K \setminus \{k\}} \exp\left(\mathbf{z}_k \cdot \mathbf{z}_j/\tau\right)}, \tag{7}$$

where $p(k)$ is the index of the data paired with the data of index $k$ with the same intent. We mix the new data from the non-experts with a subset of the original training data to update the Dif-VAR, producing a more accurate reward function by Eq. 4. The robot can then self-improve its policy network with the reward function by randomly sampling a sound command as the goal. The collection of the visual-audio pairs does not require special equipment other than an RGB camera and a microphone. The users provide images and sound based on their common knowledge using their own voices. The users do not need to type in speech transcriptions, draw bounding boxes and

masks, modify the network architectures, or design a reward function. To fine-tune VAR in [1], non-experts have to provide a sound command with different intent $\mathbf{S}^-$ for each image $\mathbf{I}$ to use triplet loss. In contrast, Dif-VAR eliminates this requirement by utilizing the SSC, leading to a more intuitive data collection experience for non-experts and better performance with fewer labeled data. See Appendix A for the fine-tuning algorithm.

## 4  Experiments

In this section, we first describe the various environments and sound datasets. Then, we compare the performance and data efficiency of our pipeline with several baselines and ablation models.

### 4.1  Environments and sound dataset

**Robotic environments:** We evaluate the performance of all the methods on three different robotic environments: iTHOR, Desk, and Row. In all environments, the perception of the robot comes from a monocular uncalibrated RGB camera, and the robot must fulfill the sound command. See Appendix B for details.

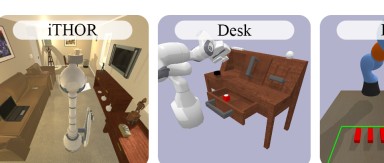

Figure 3: Simulation environments.

**Sound data:** We use several types of sounds from state-of-the-art datasets in training and testing. Specifically, we use speech signals from Fluent Speech Commands (FSC) [16] and short speech commands from Google Speech Commands (GSC) [44]. We also collect a synthetic speech dataset using Google Text-to-Speech. We use single-tone signals from NSynth [45] and environmental sounds from UrbanSound8K (US8K) [46] and ESC-50 [47]. The *Wordset dataset* was created from the "0," "1," "2," "3" in GSC. We also used a *Mix dataset* to show that the Dif-VAR can map multiple types of sounds to a single object or idea, by mixing speech data with environmental sound. See Appendix C for more sound examples and intent we choose for the environment.

### 4.2  Evaluation of the RL policy

**Evaluation metrics.** We evaluate the model with two metrics: (1) success rate (SR) and (2) the number of labels used for training (LU). We define SR as the percentage of successful test episodes. We test the learned policy for 50 episodes for each intent across multiple random seeds, and an agent succeeds if it fulfills the command. We compare the label usage of the models because a command following robot deployed in the real world should require as few annotations as possible from non-experts for fine-tuning.

**Baselines and ablations.** We compare the RL performance of our method against the following baselines and ablations. (1) "E2E" is a representative end-to-end deep RL policy for voice-controlled robots [7]. E2E uses hand-tuned task-specific reward functions and requires ground-truth class labels for image and sound classification. (2) "VAR" trains an RL agent based on the output of the VAR [1]. VAR utilizes triplet loss for training and fine-tuning. Both our method and VAR use Eq 5 for the downstream RL tasks. (3) We compare the performance between flat (F) and hierarchical (H) architectures that have the same total trainable parameters. (4) "ASR+NLU+RL (ANR)" is a common modular pipeline. Note that, unlike this baseline, our method does not rely on any transcriptions or expertise to be fine-tuned. This baseline does not work with non-speech datasets such as NSynth, which will be indicated by "-". (5) "CLIP" uses ASR for speech recognition and uses the dot product of the embeddings from the CLIP model as the reward [48]. We use "CLIP" as a representative of pre-trained visual-language models that claim zero-shot transferability to downstream tasks [48]. (6) "Oracle" is an RL agent which assumes perfect ASR and NLU modules and is trained with hand-tuned reward functions and ground-truth class labels. See Appendix E for more details.

**Definition of labels.** In this paper, labels include all forms of annotation and measurement that are used to train a model. For example, one-hot labels for image and sound classification and the

Table 1: Test success rate with various types of sound commands in the original visual domain.

| Env | Steps $(\times 10^6)$ | Dataset | SR↑ | | | | | | |
|---|---|---|---|---|---|---|---|---|---|
| | | | CLIP | ANR | E2E | VAR | Ours(F) | Ours(H) | Oracle |
| Row | 3.0 | Wordset | 1.5 | 85.5 | 95.5 | 97.0 | **98.0** | 96.0 | 98.0 |
| | 3.0 | NSynth | - | - | 92.5 | **98.0** | **98.0** | 97.0 | 98.0 |
| | 3.0 | Mix | - | - | 94.0 | 95.5 | **97.0** | 95.0 | 98.0 |
| Desk | 9.0 | Mix | - | - | 77.0 | 58.5 | 84.5 | **89.5** | 90.0 |
| iTHOR | 9.0 | FSC | 10.8 | 66.0 | 68.0 | 65.6 | 72.4 | **76.8** | 79.2 |

distance measurement between the robot and the goal are both labels. One visual-audio pair $(\mathbf{I}, \mathbf{S}, y)$ for training or $(\mathbf{I}, \mathbf{S})$ for fine-tuning used in Dif-VAR requires 1 label to indicate $y$ or the same intent. A visual-audio triplet used in VAR, $(\mathbf{I}, \mathbf{S}^+, \mathbf{S}^-)$, requires 2 labels to indicate the positive and the negative. Every E2E training step requires about 3 labels, including the target object state checking (e.g. check if the light is switched on), distance measuring to calculate the extrinsic reward, and a one-hot label for auxiliary losses.

**Control policies with unheard sounds.** In this experiment, we test the performance of different models with sound commands never heard by the agent during training (e.g. new speakers). All the models are trained with the same number of RL steps and sufficient labels. They are tested in the original Floor Plans 201-220 or desk. No fine-tuning is performed yet.

Table 1 suggests that the intrinsic rewards produced by our representation adequately support the RL training across various robots, robotic tasks, and types of sound signals. Remarkably, our method demonstrates satisfactory performance even without the inclusion of extrinsic rewards. CLIP suffers from a severe domain shift problem in our task. As a general-purpose pretrained model, CLIP is not tailored to generate an RL reward or designed for any specific downstream robotic applications. ANR is limited to speech signals and has lower SR than the other methods due to intermediate errors, which coincides with the findings in [1, 22]. Compared to our method, the VAR does not perform well in every environment, which suggests that the Dif-VAR produces better representation and more reliable rewards. The flat architecture outperforms the hierarchical architecture in only the Row environment, indicating that the hierarchical architecture is more suitable when the skills required to complete tasks vary. See Appendix D for examples of task execution of the agent.

**Fine-tuning in novel domains.** This experiment aims to show the potential of each method to be improved in a new domain. We consider the scenario where a trained household robot is purchased to serve in a new place. Each method is given the same number of new labels, and a data-efficient method should achieve the highest success rate. We first test the performance of trained models with unheard sound commands in unseen domains with-

Table 2: Average success rates over unseen domains before and after fine-tuning.

| | iTHOR(sim) | | Desk(sim) | | Row(real) | |
|---|---|---|---|---|---|---|
| **LU** | 0 | 253 | 0 | 150 | 0 | 300 |
| ANR | 18.8 | 19.8 | - | - | 13.8 | 15.0 |
| E2E | 18.4 | 19.8 | 44.5 | 45.0 | 15.0 | 15.0 |
| VAR | 19.6 | 57.9 | 35.5 | 58.0 | **18.8** | 56.3 |
| Ours(F) | 20.8 | 85.8 | 69.0 | 84.5 | **18.8** | **78.8** |
| Ours(H) | **23.2** | **88.4** | **70.0** | **90.0** | 16.3 | 75.0 |

out any fine-tuning. For the iTHOR environment, the agent is tested in Floor Plans 226-230, which have sets of furniture and arrangements that are unfamiliar to the robot. For the Desk environment, the robot is placed in front of a new desk with unseen object appearances and locations. The results are marked by "LU=0" in Table 2 as 0 new label is required for the test. We see that the performance of all methods drops compared to the test results in the original domains shown in Table 1. This phenomenon is due to the common problem of domain shift faced by learning systems [8]. We then manually collect an average of 253 new labels for each unseen floor plan to fine-tune each method for that floor plan in the iTHOR environment, and 150 new labels for the unseen desk in the Desk environment. We followed Sec. 3.3 to fine-tune the VAR and Dif-VAR and used Eq. 4 to self-improve RL policies without current sounds for 1M timesteps. For E2E, we collect one-hot

labels and use simulator queries during the fine-tuning. The fine-tuning is terminated after it reaches the label limit. See Appendix D.4 and D.5 for task execution before and after the fine-tuning.

From Table 2, we find that the ANR and E2E can only be improved by less than 1.5%, suggesting the inefficiency of fine-tuning these methods after deployment. The label quotas are depleted rapidly due to the inefficient use of labels for policy network fine-tuning, which leads to less RL experience. Our methods have higher data efficiency as labels were purely used to update the Dif-VAR, and there was no label consumption during the self-supervised RL exploration. This leads to an overall richer RL experience. Compared to VAR, our method achieves better performance using the same number of labels because Dif-VAR does not need negative pairs for fine-tuning. Using around 250 image-sound pairs, our method successfully improves itself to fulfill $4 \sim 5$ tasks in a new domain. We believe the effort is manageable by a non-expert. See Appendix E.6 for intermediate results versus the label usage during fine-tuning.

**Fine-tuning in the real world.** This experiment shows that our method is practical and helps minimize the sim2real gap. In this experiment, the agent performs a grasping task with a noisy background using a single un-calibrated RGB camera. This setting

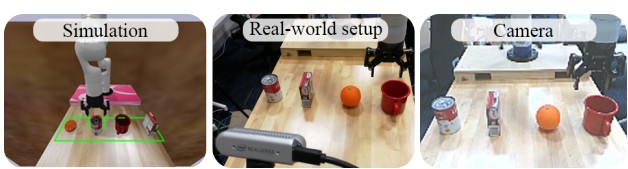

Figure 4: Sim2real experiment setup.

is challenging for user-involved sim2real because the performance of the models is sensitive to the inevitable inconsistency in the camera pose between simulation and the real world [1], and it is unlikely that non-experts know how to calibrate a camera. To solve the problem, the Dif-VAR learns to associate valid pre-grasp poses with images and speech commands. Dif-VAR outputs a high reward when the robot reaches the desired object with a correct grasping pose. We first train the agents with domain randomization [8] in the simulator. Then, we deploy the model to a real Kinova-Gen3 arm and perform 20 tests for each intent (80 in total). See Fig. 4 and Appendix B.2 for more details. We spent an hour collecting the visual-audio pairs and fine-tuning the policy. The last two columns of Table 2 shows that our methods minimized the domain shift with a reasonable number of newly provided pairs. Qualitative results are shown in Fig. 5 and the supplementary video.

Sound: "Would you mind giving me the fourth object", Target: Pick up the rightmost object

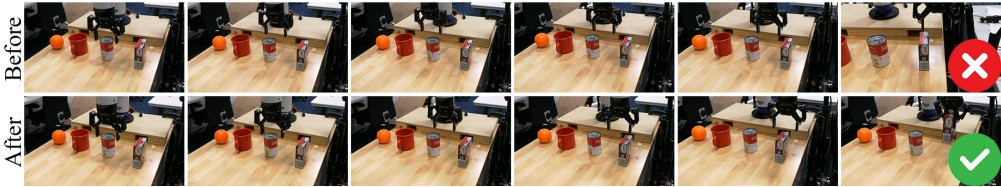

Figure 5: Before and after the fine-tuning in the real world.

## 5   Conclusion, Limitations and Future work

In conclusion, we propose a novel visual-audio representation named Dif-VAR for command following robots based on the recent advancement in (self-)supervised contrastive learning. Dif-VAR requires much fewer labels from non-experts during fine-tuning but produces higher-quality rewards for downstream RL agents. Our results suggest that visual-language association and skill development are highly correlated and thus need to be designed together. Furthermore, we are the first to demonstrate that (self-)supervised contrastive loss has the potential to enhance human-robot interaction. However, our work has the following limitations, which open up directions for future work. First, empty intents may result in sparse intrinsic reward functions, which pose challenges in long-horizon tasks. Our reward function can be combined with other intrinsic rewards [49]. To increase robustness, we may take 3D information and history into reward calculation. Second, user studies are needed to confirm that collecting visual-audio pairs is intuitive and convenient for end-users. Third, our method can be combined with imitation learning to further improve data efficiency.

**Acknowledgments**

This work is supported by AIFARMS through the Agriculture and Food Research Initiative (AFRI) grant no. 2020-67021-32799/project accession no.1024178 from the USDA National Institute of Food and Agriculture. We thank Yunzhu Li and Karen Livescu for insightful discussions and all reviewers for their feedback.

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

# A  Algorithm for fine-tuning an agent

---

**Algorithm 1** Fine-tuning the Dif-VAR (F) and an RL agent

---

1: Inputs: A trained Dif-VAR $\mathbf{V}$, a trained policy $\pi_\theta$ , a subset of the original training data $\mathcal{D}_{old}$
2: Collect a small set of visual-audio pairs $\mathcal{D} = \{(\mathbf{I}_i, \mathbf{S}_i)\}_{i=1}^U$
3: $\mathcal{D}_{new} = \mathcal{D}_{old} \bigcup \mathcal{D}$
4: **for** a sampled minibatch $\{(\mathbf{I}_i, \mathbf{S}_i)\}_{i=1}^N$ from $\mathcal{D}_{new}$ **do**                    ▷ Fine-tune Dif-VAR
5:        Calculate empty intent label $e_i$ by checking if $\mathbf{S}_i = \mathbf{0}_{l \times m}$
6:        Calculate image and sound embeddings: $\mathbf{h}^I, \mathbf{z}^I, \mathbf{h}^S, \mathbf{z}^S \leftarrow \mathbf{V}(\mathbf{I}_i, \mathbf{S}_i)$
7:        Calculate $\mathcal{L}_{\text{SSC}}$ by Eq. 7
8:        Calculate loss by $\mathcal{L}_{\text{finetune}} = \alpha_1 \mathcal{L}_{\text{SSC}} + \alpha_2 \frac{1}{N} \sum_{j=1}^N \mathcal{L}_{\text{BCE}}(b^I(\mathbf{h}_j^I), e_j)$
9:        Update $\mathbf{V}$ to minimize $\mathcal{L}_{\text{finetune}}$
10: **for** $k = 0, 1, 2, ...$ **do**                    ▷ Self-supervised RL fine-tuning
11:        Sample a sound command $\mathbf{S}_g$ from $\mathcal{D}$ as goal
12:        **for** $t = 0, 1, ..., T$ **do**
13:            Receive RGB image $\mathbf{I}_t$ and robot state $\mathbf{M}_t$
14:            Calculate image and sound embeddings: $\mathbf{v}_t^I, \mathbf{v}_g^S \leftarrow \mathbf{V}(\mathbf{I}_t, \mathbf{S}_g)$ by Eq. 3
15:            Calculate reward $r_t = \mathbf{v}_t^I \cdot \mathbf{v}_g^S$
16:            **if** $\mathbf{S}_t$ **then**
17:                Calculate embeddings: $\mathbf{v}_t^S \leftarrow \mathbf{V}(\mathbf{S}_t)$
18:                $r_t = r_t + \mathbf{v}_t^S \cdot \mathbf{v}_g^S$
19:            Store $\{r_t, \mathbf{I}_t, \mathbf{M}_t, \mathbf{v}_t^I, \mathbf{v}_g^S\}$ in a memory buffer $\mathcal{D}_{RL}$
20:        Update $\pi_\theta$ with data from $\mathcal{D}_{RL}$ using PPO
21:        Clear $\mathcal{D}_{RL}$
22: **return** $\mathbf{V}, \pi_\theta$

---

# B  Robotic environment descriptions

The Row and Desk environments are developed in PyBullet [50] and focus mainly on manipulation tasks. In contrast, the iTHOR environment is developed in AI2-THOR [51] and is challenging in perception and designed for mobile robots.

## B.1  Row

Four objects are placed in a line at a random location unknown to the robot on the table. A robot arm needs to move its gripper and stay above the object corresponding to a given command based on RGB images. The camera is placed at a fixed location on the side of the table such that it can capture the gripper and the objects from a distorted perspective. The relative positions of the gripper tip and the objects are initialized randomly at the beginning of an episode. A sound command only mentions the orindal information about the target object, and the robot needs to develop spatial reasoning skills to approach the target object using the relative positional information observed from the camera.

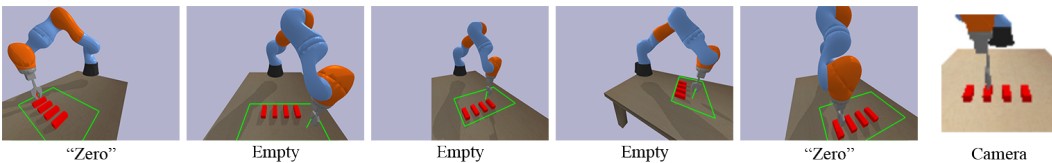

Figure 6: Visualization of the Row environment using Kuka-iiwa robot arm with paired images and voices from the Wordset. In this case, "zero" means the leftmost block, "one" means the second block from the left, and so on. The red and green rays are just for illustration purposes. The possible locations of the blocks are limited to the green rectangle and the end-effector location is indicated by the vertical ray. The rightmost figure shows the camera view.

## B.2  Row - real

This environment is modified from the original Row environment. The Four objects are a mug, a soup can, a pudding box, and an orange. Different objects may require distinct grasping poses. See Fig. 7 for examples. At the end of an episode, the gripper performs a grasp by lowering its height from its current position, closing the fingers, and lifting the object up. For domain randomization, we randomize the background, camera viewpoint, and relative offset among the objects.

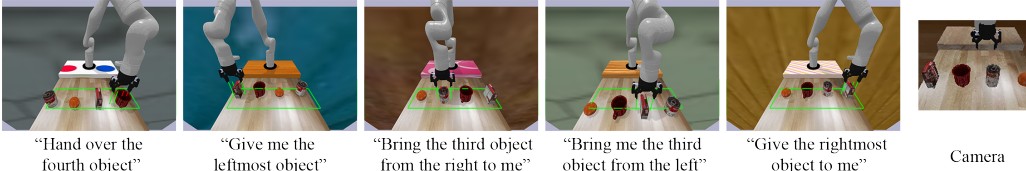

"Hand over the fourth object"  "Give me the leftmost object"  "Bring the third object from the right to me"  "Bring me the third object from the left"  "Give the rightmost object to me"  Camera

Figure 7: Visualization of the Row environment with paired images and voices from the Synthetic dataset under domain randomization setting. The grasping poses for the mug and the pudding box are different. The red and green rays are just for illustration purposes. The rightmost figure shows the camera view.

## B.3  Desk

The Desk environment is modified from the CALVIN dataset [52]. A Franka Panda robot arm is placed in front of a desk with a sliding door and a drawer that can be opened and closed. On the desk, there is a button connected to an alarm clock, a switch to control a light bulb, and a pill case. The tasks of the robot include turning on or off the light bulb by manipulating the switch, pressing the button to mute the alarm clock and turn the LED of the clock into red, and picking up the pill case that could be on the top of the desk or inside a closed drawer. When the pill case is located inside a closed drawer, the robot needs to open the drawer before picking up the pill case. The sound commands come from FSC, ESC-50, and the Synthetic dataset.

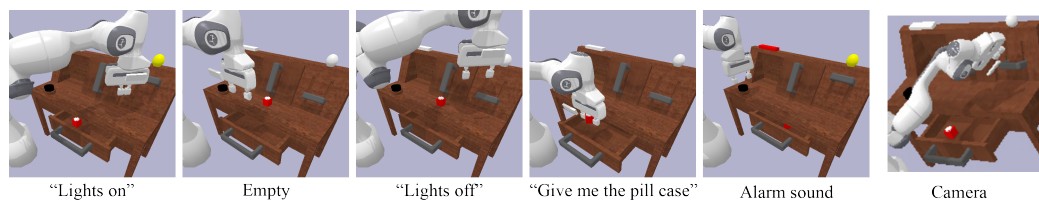

"Lights on"  Empty  "Lights off"  "Give me the pill case"  Alarm sound  Camera

Figure 8: Visualization of the Desk environment with paired images and voices. The rightmost figure shows the camera view.

## B.4  iTHOR

Our iTHOR environment uses real full-sentence speech commands to simulate a real-world application of household robots. The environment has 30 different floor plans of living rooms, each with its own set of decorations, furniture, and arrangements. The robot is given goal tasks such as switching the floor lamp or television on or off. The robot must navigate through the environment and interact with the intended object given RGB images and a noisy local discrete occupancy grid as the robot states. The complexity of the environment requires the agent to associate complicated speech commands with high-fidelity visual observations, without a floor plan map. The floor plans can be visualized and interacted with in https://ai2thor.allenai.org/demo/.

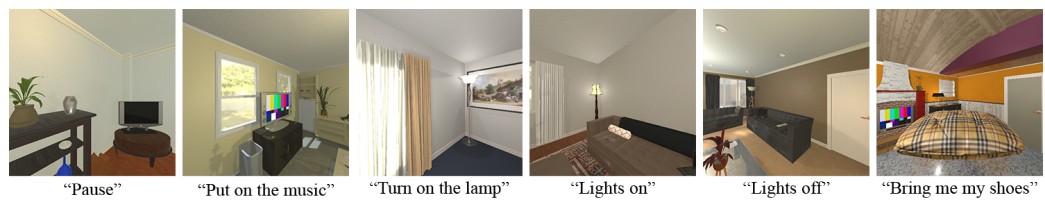

"Pause"  "Put on the music"  "Turn on the lamp"  "Lights on"  "Lights off"  "Bring me my shoes"

Figure 9: Visualization of the iTHOR environment with paired images and voices from the FSC dataset.

## B.5 List of Tasks

Table 3: List of tasks in each environment.

| Envs | Tasks in the original domain | Changes in the new domain |
|------|------------------------------|---------------------------|
| iTHOR | activate the floor lamp
deactivate the floor lamp
activate the TV
deactivate the TV
find the pillow | unseen furniture, room decoration, room arrangement, and voices from new speakers |
| Desk | activate the light bulb
deactivate the light blub
mute the alarm clock
pick up the pill case | unseen desk, object locations, object appearance, and sound from new speakers or the alarms |
| Row | first block
second block
third block
forth block | unseen sound or voices |
| Row - real | first object
second object
third object
forth object | unseen camera intrinsics, camera extrinsics, background, relative locations among the objects, and voices from new speakers |

## C  Sound Data

Table 4: Sound signals used in the experiments.

| Dataset | Sound | Examples |
|---------|-------|----------|
| FSC | activate light
deactivate light
activate music
deactivate music
bring shoes | "Turn on the lights," "Lamp on"
"Switch off the lamp," "Lights off"
"Put on the music," "Play"
"Pause music," "Stop"
"Get me my shoes," "Bring shoes" |
| GSC | "0," "1," "2," "3"
names of 4 objects | "zero," "one," "two," "three"
"house" "tree," "bird," "dog" |
| NSynth | $C_4$, $D_4$, $E_4$, $F_4$ | Various instruments, tempo, and volume |
| US8K | bark, jackhammer | Sound recorded in the wild |
| ESC-50 | Clock alarm | Alarm sound emitted from various alarm clocks |
| Synthetic | bring pill case
first object

second object

third object

fourth object | "Pass over the pill box for me," "Give me the pill case"
"I would like the first object," "Give me the leftmost object"
"Would you mind giving me the second object from the left,"
"Bring the third object from the right to me"
"Take the third object," "Bring me the second object from the right"
"Take the third object from the left"
"Give the rightmost object to me ," "Hand over the fourth object" |

# D    Visualization of task execution

## D.1    Row

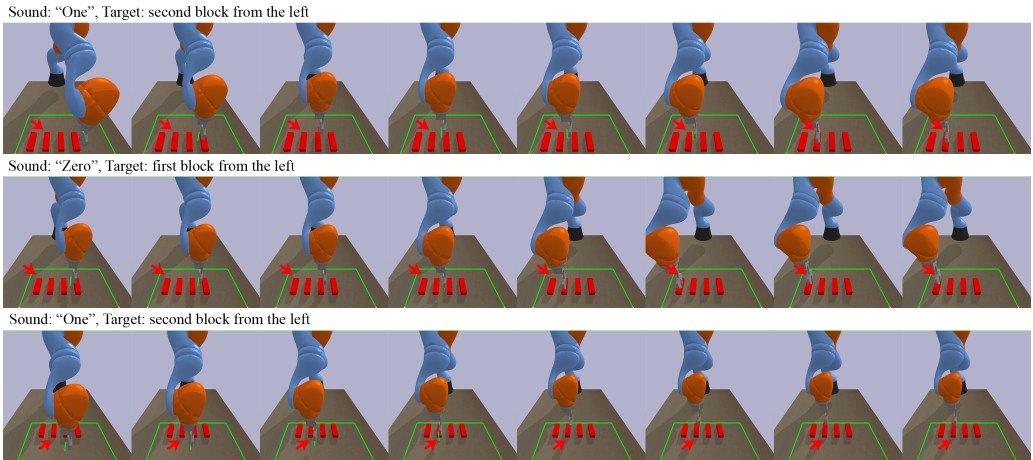

Figure 10: Visualization of the task execution in the Row environment after training without fine-tuning. The sounds come from Wordset dataset. Kuka moves its gripper to the target block successfully in all episodes.

## D.2    Desk

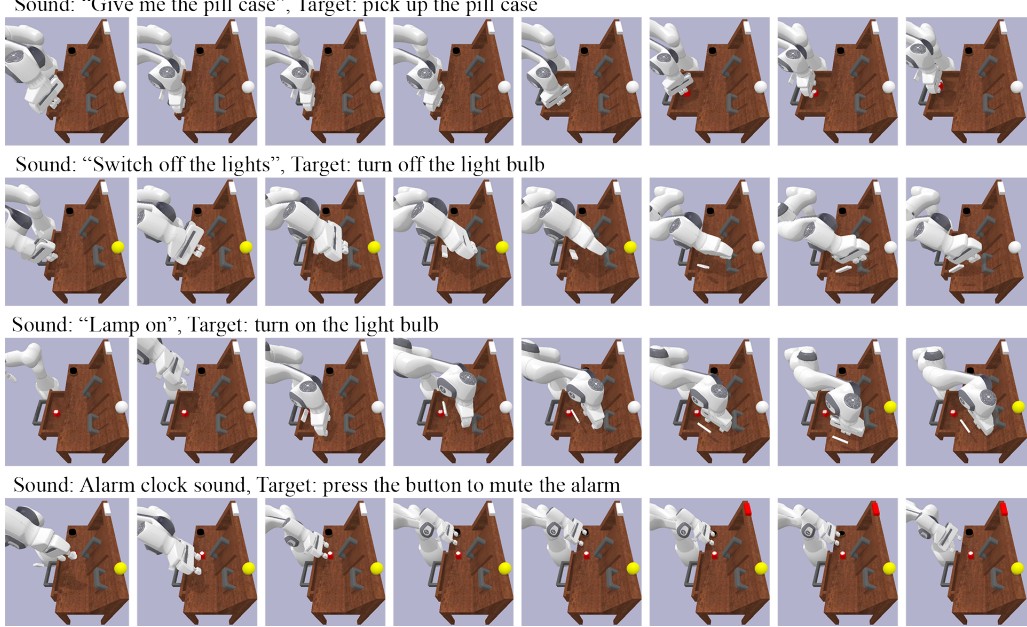

Figure 11: Visualization of the task execution in the Desk environment after training without fine-tuning.

## D.3 iTHOR

Sound: "Bring me my shoes", Target: pick up the pillow

Sound: "Switch off the lights", Target: turn off the floor lamp

Sound: "Play", Target: turn on the TV

Sound: "Stop the music", Target: turn off the TV

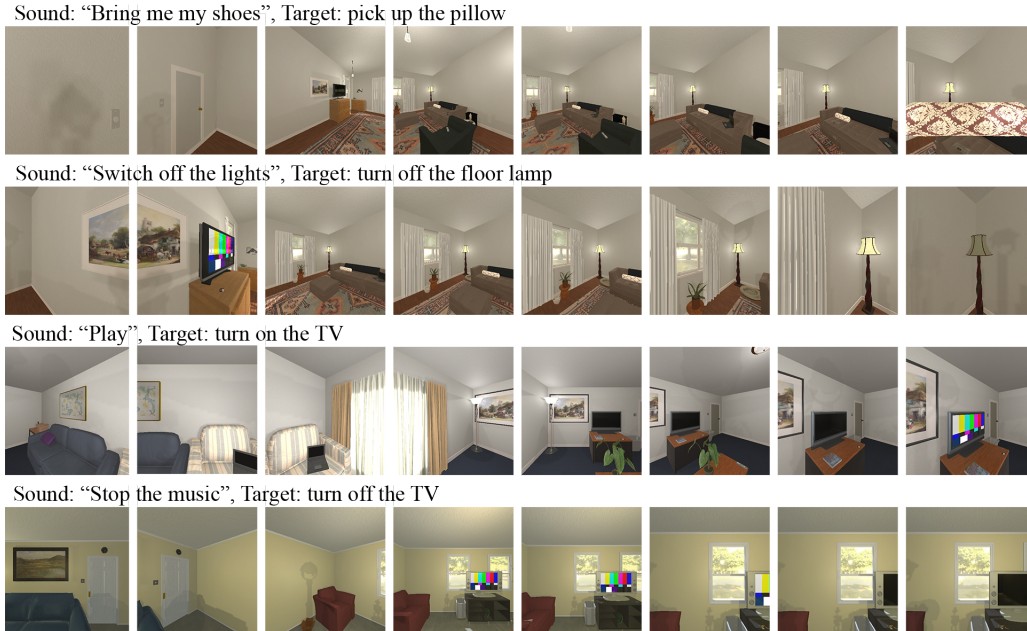

Figure 12: Visualization of the task execution in the iTHOR environment after training without fine-tuning. The sounds come from FSC dataset. iTHOR agent finishes household tasks successfully in all episodes.

## D.4 iTHOR fine-tuning

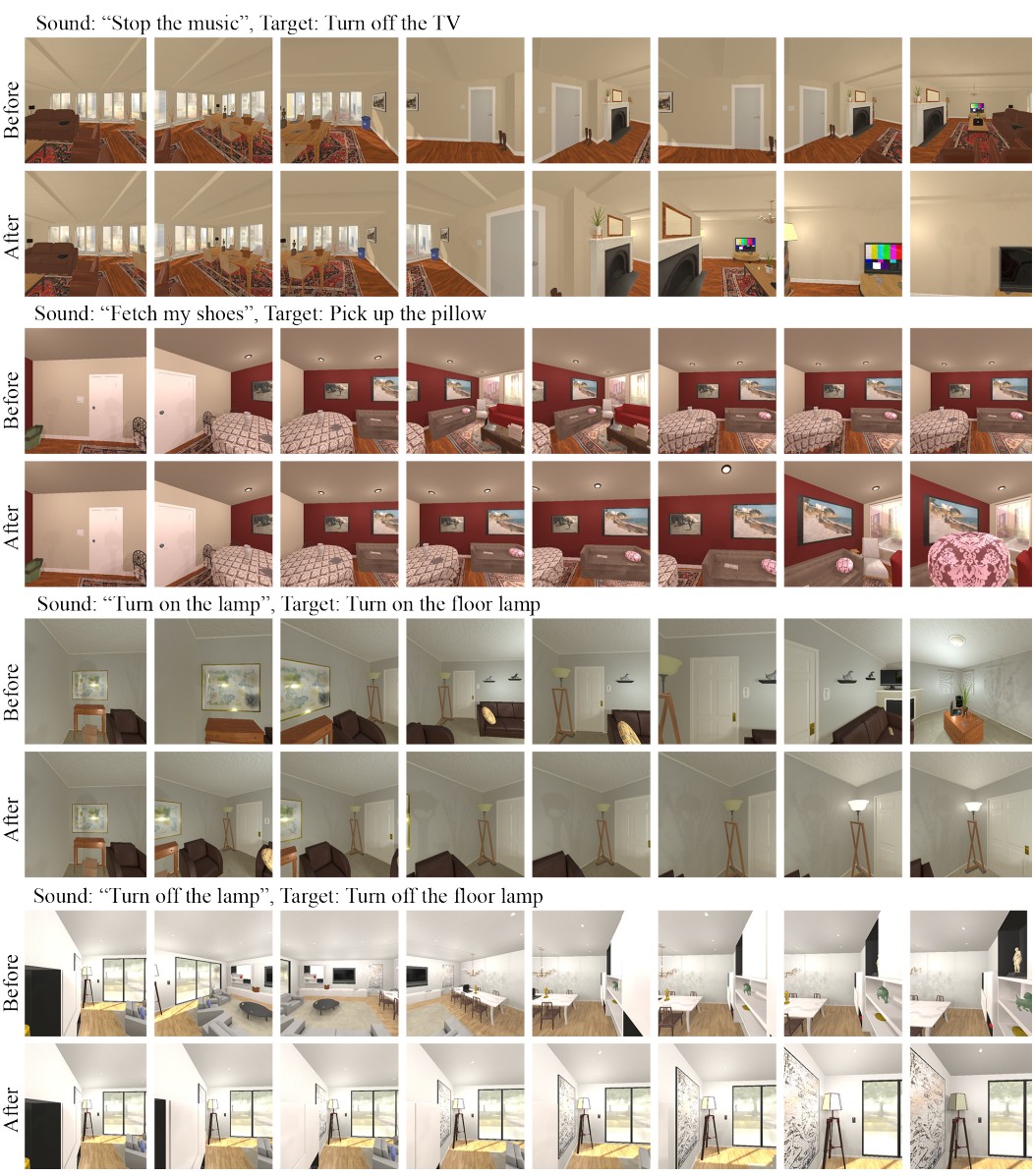

Figure 13: Visualization of the task execution in the iTHOR environment before and after the fine-tuning in unseen floor plans and the sound commands given by new speakers.

## D.5 Desk fine-tuning

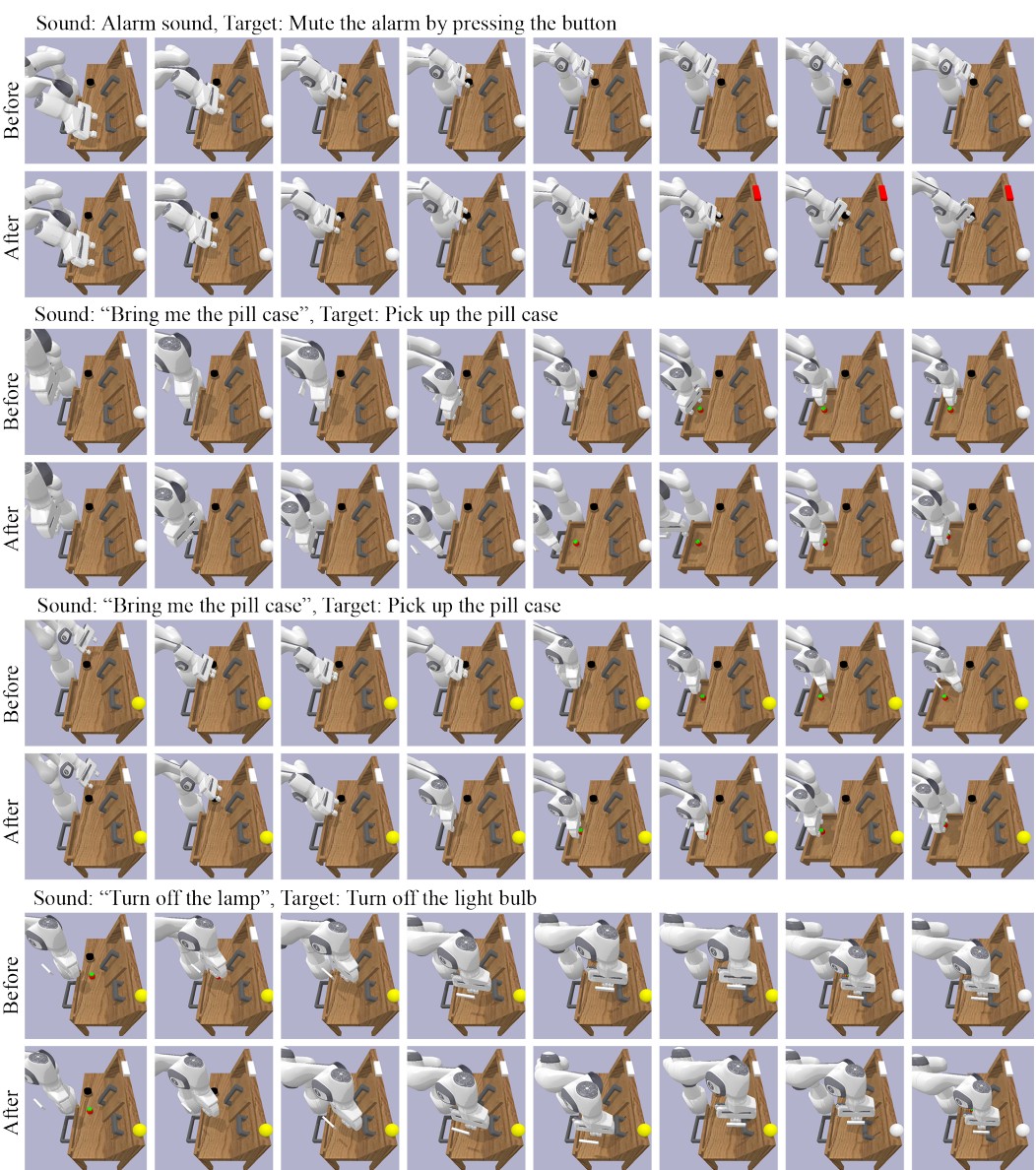

Figure 14: Visualization of the task execution in the Desk environment before and after the fine-tuning with a unseen desk and the sound commands given by new speakers. The appearance of the desk and the pill case are different from the original desk. The location of the light bulb, the button, the LED, and the drawer are different from the original desk.

# E    Facts and details

## E.1    Time efficiency

We evaluate the time efficiency of all the methods. All the models are running on a single Nvidia GTX 1080 Ti GPU and a Intel(R) Core(TM) i7-8700 CPU @ 3.20GHz. We report the average time in second (s) for the model to take one action in the iTHOR environment with the FSC dataset. The average is calculated from 12500 samples.

- ANR: 0.041s
- E2E: 0.018s
- VAR: 0.024s
- Dif-VAR: 0.022s

## E.2    ANR

- Implementation: We first use an off-the-shelf automatic speech recognition (ASR) named Mozilla DeepSpeech [53] to transcribe the speech to text. We then train a learning-based natural language understanding (NLU) module to handle the noisy output from the ASR [54]. For example, "Play the music" is sometimes transcribed as "by the music." Finally, a vision-based RL agent operates with the predicted intent from the NLU.
- Accuracy of intent prediction of ASR+NLU: FSC dataset: 86.0%; Wordset: 87.0%.
- Fine-tuning details: We fine-tune the RL agent of this pipeline because it is the major source of performance degradation. The policy network is initialized with the weights obtained during the training. During the fine-tuning, the input of the model includes images and ground-truth intent IDs as if the ASR+NLU is perfect. The model also receives a one-hot label of the images and reward signals from the simulator as supervision signals. The policy network is updated based on the PPO loss and the auxiliary loss for the images.

## E.3    E2E

Fine-tuning details: We fine-tune the whole policy network end-to-end. The policy network is initialized with the weights obtained during the training. During the fine-tuning, the input of the model includes images and raw sound signals. The model also receives a one-hot label of the images, ground-truth intent IDs, and reward signals from the simulator as supervision signals. The policy network is updated based on the PPO loss and the auxiliary losses for the images and audio.

## E.4    VAR

Fine-tuning details: We fine-tune the VAR and let the VAR provide rewards and observations to fine-tune the policy network. Both VAR and the policy network are initialized with the weights obtained during the training. We collect visual-audio triplets consisting of an image, positive audio, and negative audio from the environment and fine-tune the VAR using the triplet loss. During the fine-tuning of the policy network, the input includes images and vector embeddings from the VAR. The policy network is updated based on the PPO loss.

### E.5 Dif-VAR

We show the statistics of training and fine-tuning data for Dif-VAR.

Table 5: Number of training and fine-tuning pairs

| Envs | # of training pairs | # of fine-tuning pairs | Ratio (Fine-tuning : training) |
|---|---|---|---|
| Row - real | $5.0 \times 10^4$ | $3.0 \times 10^2$ | 0.6% |
| Desk | $2.5 \times 10^4$ | $1.5 \times 10^2$ | 0.6% |
| iTHOR | $6.0 \times 10^4$ | $2.5 \times 10^2$ | 0.4% |

We report the change in success rate w.r.t. (1) label usage for fine-tuning Dif-VAR, and (2) number of RL steps given the fine-tuned representations with the specific number of labels.

Table 6: Success rate with varying label usage and RL fine-tuning steps for Ours(F) in the Desk and iTHOR environments.

| Envs | Episode Length | LU | RL Steps | | |
|---|---|---|---|---|---|
| | | | 0.1M | 0.5M | 1.0M |
| Desk | 100 | 50 | 77.5 | 79.5 | 79.5 |
| | | 100 | 78.0 | 80.0 | 84.0 |
| | | 150 | 78.0 | 82.0 | 84.5 |
| iTHOR | 50 | 80 | 39.4 | 45.3 | 56.3 |
| | | 160 | 45.7 | 56.7 | 68.4 |
| | | 250 | 58.6 | 77.5 | 85.8 |

Table 7: Success rate with RL fine-tuning steps for Ours(F) in the Row-real environments.

| Envs | Episode Length | LU | RL Steps | |
|---|---|---|---|---|
| | | | 1100 | 2200 |
| Row-real | 20 | 300 | 40.0 | 77.5 |

## E.6 Intermediate fine-tuning performance

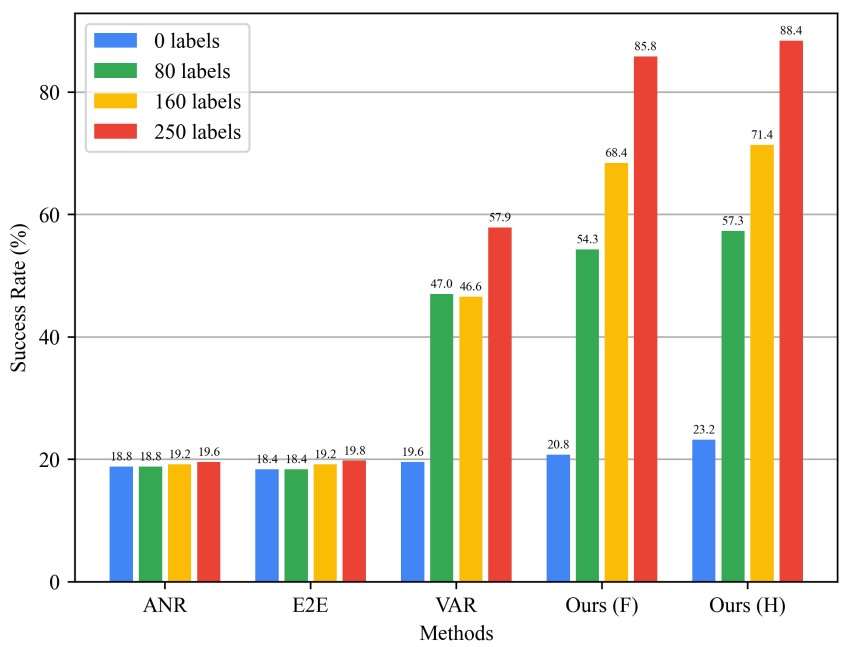

Figure 15: Success rate with varying number of labels in the iTHOR environment.

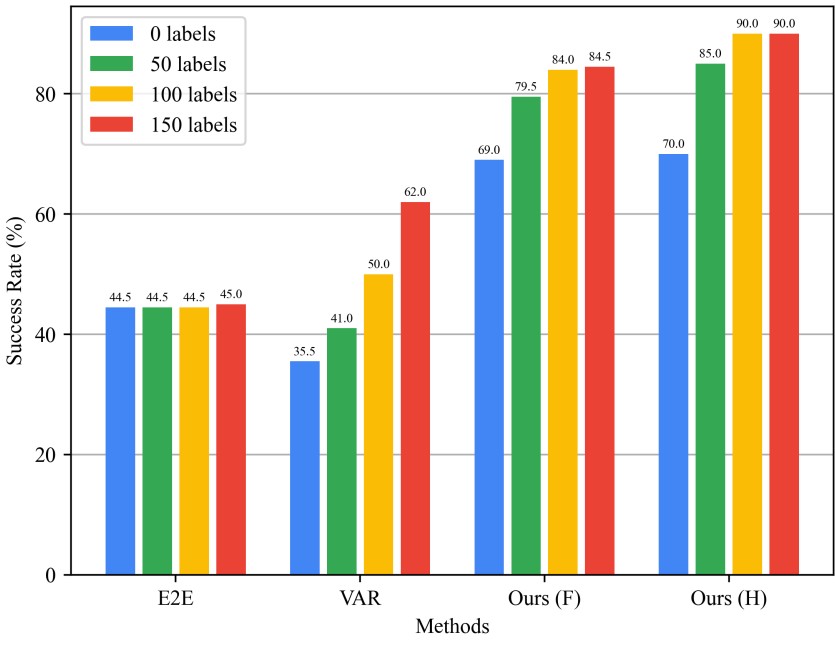

Figure 16: Success rate with varying number of labels in the Desk environment.

# F Qualitative comparison of the representation

We visualize the VARs by projecting images and sounds to the joint space, as shown in Fig. 17. We see that the embeddings of the same concept form a cluster and all clusters are separated from each other. Compared to VAR, the clusters in Dif-VAR have better intra-cluster cohesion and inter-cluster separation, suggesting that the two distinct concepts are better distinguished and the same concepts are better related. During fine-tuning, although Dif-VAR does not have $\mathbf{S}^-$ as an explicit indication of negatives like VAR does in the input, Dif-VAR can still maintain relatively clear inter-cluster separation and provide reliable rewards for the self-improvement of RL agents.

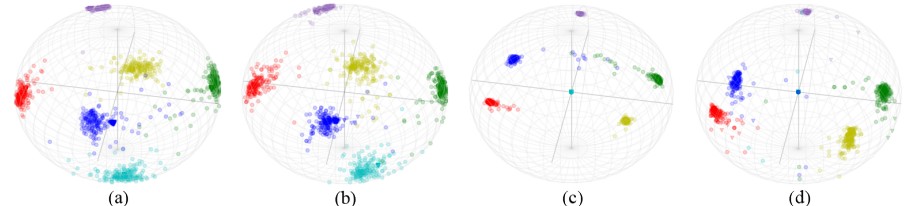

Figure 17: **Visualizations of the VARs in the iTHOR environments with FSC.** The colors indicate the ground truth intent ID of embeddings of sound (marked by triangles) and image (marked by circles). (a) VAR after the training. (b) VAR after the fine-tuning. (c) Dif-VAR after the training. (d) Dif-VAR after the fine-tuning.

# G Illustration of data collection

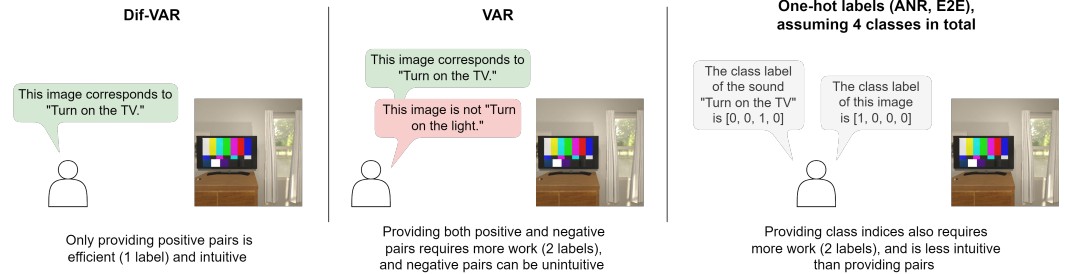

Figure 18: **Comparison among different data collection methods.** *Left*: Our method only asks for pairing an image with an audio without the need for a class label. *Middle*: VAR requires an additional pairing process than our method. *Right*: ANR and E2E require two assignments of the underlying class label which is not intuitive and need more effort.

