# OpenReview forum: "A Data-Efficient Visual-Audio Representation with Intuitive Fine-tuning for Voice-Controlled Robots"
_robot-learning.org/CoRL/2023/Conference — CoRL 2023 Poster_

### Official Review · Reviewer_TXeX · 2023-07-18

**Confidence:** 3
**Originality:** Good
**Technical Quality:** Fair
**Clarity Of Presentation:** Very Good
**Impact:** 3

**Recommendation:**

Weak Reject: I recommend rejecting the paper, but will not argue for my recommendation if the majority of other reviewers have a different opinion.

**Review:**

Strengths:
- It addresses the issue of continual learning in new environments, which is a relevant problem to the real-world deployment of learning robots.
- It attempts to mitigate the cost of acquiring datasets for training and fine tuning, which is always a concern for learned robotics.
- The proposed system is meaningfully novel.

Weaknesses:
- The paper does not adequately support its claim that this method of data labeling is easier or quicker than other methods.
- The definition of labels given may be overly generous to the method described here, in which an audio-visual pair is described as a single label rather than two labels. If this is expressed as two labels rather than one, the results would be much different compared to other methods.
- Real-world experimentation is limited to a simple grasping task.
- This work is not relevant outside the realm of audio-commanded robotics, and within that realm it is limited enough in its application that I don't believe it is likely to see further use.

**Quality Of The Limitations Section:**

Additional details required

**Questions For Rebuttal:**

1. I’m not entirely clear on the benefit of using direct sound processing over translating sound into text first. As this is central to the novelty of this paper, I recommend expanding upon it.
2. Under “Fine-tuning in novel domains”, I’d like to see more data that shows model improvement during fine tuning over fine tuning episodes collected, not just labels used. The label use information can and should be included here, but having both would provide a clearer picture.
3. In the data given in Table 2, the number of labels used to demonstrate improvement in fine tuning seem to be arbitrarily selected- 253, 150, 300. Could you provide more clarification on why these are the label limits for each task? Perhaps intermediate results could be shown, e.g. finetuning improvement for each 100 additional labels?
4. Your real-world fine tuning evaluation is based on a very low number of episodes- only 40 episodes total were used to evaluate your fine tuned model in the real world, if I'm understanding the wording on line 308 correctly. At that low number, the result could be skewed by more than +- 2% based on a single additional episode. Given the inherent unpredictability of learned robotic systems, that variance could add up quickly, for better or for worse. Would it be possible to perform more experiments for finer grained results?
5.  The "definition of labels" given may be overly generous to the method described here, in which an audio-visual pair is described as using only a single label. Why are image and sound not counted as independent labels, given that they are two separate pieces of data? Could the same logic not be applied to other methods to combine multiple pieces of data into a single label?
6. Small nit: In line 138, you refer to the latent space of the output of your indicator function as “joint space”. This is an overloaded term in robotics, as joint space could also refer to a vector whose components are the displacements of a robotic joint. You may want to consider changing that verbiage for the sake of clarity.


**Robotics Focus:**

Sufficient demonstration on hardware

**Summary Of Paper:**

This paper proposes a method of using pairs of images and sounds (e.g. spoken words, phrases, and other sounds) as labels with which to train and fine tune models used for robotic command-following tasks. These sound-image pairs are used to train DiF-VAR, which embeds these sounds and images into a network that produces unit vectors in close proximity to the related sound or image. This trained DiF-VAR is then used to create a reward function used to train an RL agent on some robotic task. Following initial deployment, fine tuning can be performed to improve performance of both the DiF-VAR and the RL policy using only sounds as labels. This system is tested in simulated and real environments. The authors claim that this system provides substantial benefit to the deployment of consumer-facing learning robotic systems due to the relative ease of fine tuning in novel environments.

**Summary Of Recommendation:**

Overall, this paper is of fair quality and is clearly written. The system shown is meaningfully novel, but I'm not sure that the system is compelling enough for this system to have an impact on future works. If the authors are able to demonstrate their claim that this system is easy to fine tune for non-technical users compared to alternatives, then I would be more confident in the utility of this work going forward. They should also address the disparity in how they count the number of labels used by their own method versus others, which could cause these results to be overly optimistic.

---

### Official Review · Reviewer_S2KB · 2023-07-19

**Confidence:** 3
**Originality:** Very Good
**Technical Quality:** Very Good
**Clarity Of Presentation:** Very Good
**Impact:** 4

**Recommendation:**

Strong Accept: I recommend accepting the paper and will argue for my recommendation even if other reviewers hold a different opinion.

**Review:**

The overall idea is novel and the paper is well written. Reward design is a pain for RL and data efficiency is a huge problem. The method proposed in this paper can solve these two problems very well.

**Quality Of The Limitations Section:**

Limitations are addressed clearly

**Questions For Rebuttal:**

1. For the real world fine tuning experiment, it will be great to see the performance can keep improving w/ RL exploration.
2. This method should be compatible w/ any RL algorithm. It will be great to see how it works w/ another RL algorithm in addition to PPO.
3. In theory, this method can also support language instead of audio. Would be interested in seeing how it can be combined w/ LLMs.

**Robotics Focus:**

Sufficient demonstration on hardware

**Summary Of Paper:**

This paper proposes a novel method for applying RL to robotics. The input of the system is audio command + visual. It has two steps. 1) using contrastive loss to train a audio+visual model to measure whether the audio and visual are close to each other. 2) using the audio+visual model to automatically calculate reward for RL training. The experimental results show improvements over existing baseline methods.

**Summary Of Recommendation:**

I recommend acceptance of this paper.

---

### Official Review · Reviewer_HeXC · 2023-07-20

**Confidence:** 3
**Originality:** Very Good
**Technical Quality:** Very Good
**Clarity Of Presentation:** Very Good
**Impact:** 3

**Recommendation:**

Weak Accept: I recommend accepting the paper, but will not argue for my recommendation if the majority of other reviewers have a different opinion.

**Review:**

Summary of Strengths

- the framework of combining video and audio to obtain a joint representation space that can be used to define an intrinsic reward, in novel to the best of my knowledge.

- the approach relies on very little domain knowledge about the environment/objects, and uses only monocular RGB image observations from an uncalibrated camera, combined with sound signals from humans.

- the experiments are on diverse scenarios ranging from manipulation to navigation, and there are some results with real robots showing that the approach works for sim2real transfer in robotic grasping.

- the paper is generally easy to follow and has sufficient details on the method for understanding.

Summary of Weaknesses/Questions

- intuitively, it is unclear to me why specifying commands in the form of sound is more novel or useful than specifying them in the corresponding language instructions (assuming sound to language mapping is near accurate, which is an implicit assumption made in the paper).

- it is unclear how general is the approach in terms different sounds from diverse people - I did not see details about diversity of new speakers in terms of tone, speed, accent etc.

- the real world results mention an hour of fine-tuning for robotic grasping. Is the approach feasible for more complex / long horizon tasks beyond grasping that would potentially require a lot more time for fine-tuning with RL in the real world?

- Some relevant comparisons are missing. How does the approach compare with prior works that learn vision-language embeddings? In my understanding the representation learning model can be replaced with a vision-language representation model and the rest of the pipeline including intrinsic reward computation can be kept as it is.

- [minor] For sample-efficient deployment in the real-world, can the approach be combined with bootstrapping via few demonstrations? Since RL in the real-world in highly sample inefficient.

**Quality Of The Limitations Section:**

Additional details required

**Questions For Rebuttal:**

Please refer to the review above.

- why is specifying commands in the form of sound is more novel or useful than specifying them in the corresponding language instructions?

- how general is the approach in terms different sounds from diverse people with differences in tone, speed, accent etc. ?

- the real world results mention an hour of fine-tuning for robotic grasping. Is the approach feasible for more complex / long horizon tasks beyond grasping that would potentially require a lot more time for fine-tuning with RL in the real world?

- Some relevant comparisons are missing. How does the approach compare with prior works that learn vision-language embeddings? In my understanding the representation learning model can be replaced with a vision-language representation model and the rest of the pipeline including intrinsic reward computation can be kept as it is.

- [minor] For sample-efficient deployment in the real-world, can the approach be combined with bootstrapping via few demonstrations? Since RL in the real-world in highly sample inefficient.

**Robotics Focus:**

Sufficient demonstration on hardware

**Summary Of Paper:**

This paper develops an approach for building command-following robots that can continually improve after deployment. The approach is based on image-audio representation learning that can be used to generate an intrinsic reward function for reinforcement learning on the robot. Experiments on simulated navigation and manipulation settings, and on real world robotic grasping, demonstrate that the approach can be used to continually improve robotic policies based on user commands, with less data compared to prior approaches.

**Summary Of Recommendation:**

I am not recommending accept because I have a few questions/concerns regarding the usefulness and feasibility of the approach for real world deployment.

--- AFTER REBUTTAL ---

I am increasing my score to weak accept after discusions because my concerns regarding more complex tasks than grasping are slightly orthogonal to the main goals of the paper - however, for the approach to be actually adopted by the community, I do believe these slightly more complex tasks are necessary to be shown as applicable for the proposed reward function.

---

### Official Review · Reviewer_b57y · 2023-07-21

**Confidence:** 4
**Originality:** Very Good
**Technical Quality:** Very Good
**Clarity Of Presentation:** Good
**Impact:** 4

**Recommendation:**

Weak Accept: I recommend accepting the paper, but will not argue for my recommendation if the majority of other reviewers have a different opinion.

**Review:**

Reward function is described as a similarity between two vectors that are learned representations of visual-language, this does not factor in actual policy success, or success rate. I don't think this is a correct formulation of reward ie, how likely is it to succeed given state and action. There is no action component, language is only intent.

The contrastive loss however is an interesting formulation. It allows for fluidity in language descriptions, but presented in audio form.

The real world results are not very strong, but cleaner studies in sim do drive a point.

**Quality Of The Limitations Section:**

Additional details required

**Questions For Rebuttal:**

1. Why embed in audio space as against language space?
2. How would you make the reward function more robust?

**Robotics Focus:**

Sufficient demonstration on hardware

**Summary Of Paper:**

The paper proposes a method for contrastive learning on image and audio pairs for robotic tasks. There is an intrinsic reward function being modeled. The policy is trained in sim and then finetuned to real environments. The paper explores the use of audio conditioned robotics policies.

**Summary Of Recommendation:**

The paper proposes a new framework for audio visual learning using contrastive self supervised loss. They've results on sim and real. The formulation is a little bit complicated and hard to understand so I think the paper could be written slightly better from a flow/explaining abbreviations perspective. The reward function + learning in sim makes the system not the most scalable, hence incremental advance. Overall the work is scientific and well executed.

---

### Official Review · Reviewer_Tifq · 2023-07-21

**Confidence:** 4
**Originality:** Good
**Technical Quality:** Good
**Clarity Of Presentation:** Very Good
**Impact:** 3

**Recommendation:**

Weak Accept: I recommend accepting the paper, but will not argue for my recommendation if the majority of other reviewers have a different opinion.

**Review:**

### Quality

Voice-controlled manipulation is a crucial and necessary competence a household robot should have and not a lot of work focus on the voice part. Most of the existing methods convert the voice command to a text command with a speech-to-text model and proceed with language models for downstream tasks. But the authors explore a more end-to-end way and propose a solid and clear pipeline to train the visual-audio representation, to train an RL agent with auto-generated rewards, and to fine-tune the representation in a new domain for further RL training. The method is clearly introduced. Simulation and real-world experiments are carried out to support and prove the effectiveness of the proposed method to some extent.

### Clarity

The structure of the paper is clear and it is easy to read.

### Originality

The representation learning part, the RL pipeline, and the fine-tuning part are quite standardized. The pipeline is also similar to VAR.

### Strength

1. Proven effectiveness of combining visual-audio representation learning with RL training in voice-controlled manipulation tasks.
2. Proven effectiveness of fine-tuning with the proposed method.
3. A solid solution to solve voice-controlled manipulation tasks.

### Weakness
1. The authors haven’t shown how the amount of fine-tuning data influences the results. The ratio between the pretraining data and fine-tuning data is also not shown. It is good to see that the fine-tuning works, but there still lacks experiments showing how “data-efficient” the proposed method is regarding data amount.
2. More details regarding the experiments are appreciated like the list of evaluated tasks, what tasks are novel, and how many data pairs are collected for pretraining. More details regarding how the baselines are fine-tuned are appreciated.

**Quality Of The Limitations Section:**

Limitations are addressed clearly

**Questions For Rebuttal:**

1. How does the performance vary given different amounts of extra fine-tuning data? Show how efficient it is. 250 pairs are still a lot of data to collect for non-expert users. If you can plot a figure showing SR(y-axis) vs the amount of data (x-axis), it would provide more insights about how data-efficient the method is.
2. When “unheard sound commands” is mentioned in the Experiment “Fine-tuning in novel domains” session, does it mean that the tasks are novel or the tasks are seen task but only the voice is unseen?
3. It will be good if more information regarding the training data can be shared in the appendix. How many data pairs are needed for the pretraining? What is the ratio between pretraining and fine-tuning data amount?
4. It will be good if more information regarding the list of tasks can be shared in the appendix. What tasks are the robot trained on? Are there novel tasks in evaluation?
5. How are the baselines fine-tuned? Do they require task labels to fine-tune?

**Robotics Focus:**

Sufficient demonstration on hardware

**Summary Of Paper:**

The authors propose a pipeline for training and fine-tuning visual-audio representations that can be exploited to generate intrinsic rewards for RL training on household voice-controlled manipulation tasks. In the representation learning stage, given (image, voice command audio, task_id) pairs, the authors use the contrastive loss to supervise the learning of visual and audio embedding, unifying the embeddings space from the two modalities, and pulling the embeddings for the same task id together. In the reinforcement learning stage, image observation and goal voice command audio are encoded and their cosine similarity is regarded as the intrinsic reward for RL training. When the robot is deployed in a new environment, by collecting (image, voice command audio) pairs and fine-tuning the representation with self-supervised contrastive loss, the model adapts to the new environment efficiently. Simulation and real-world results show the effectiveness of the method.

**Summary Of Recommendation:**

The reason for acceptance:

the authors propose to combine visual-audio representation learning and RL to train voice-controlled manipulation agent. The authors show that the proposed method outperforms baseline alternatives. In addition, the proposed method is easier and more effective to fine-tune and adapt to novel domains with data that can easily be collected by non-expert users. Solid simulation and real-world experiments are shown to support the main claim.

The reason for weak acceptance:

The pipeline is quite similar to the VAR paper except for some modifications in loss which means that the work is incremental in essence. The representation learning, RL, and fine-tuning pipeline are also quite standardized.

---

### Author Response · Authors · 2023-08-10
**Response to All Reviewers**

  **(For Reviewers `b57y`, `HeXC`, and `TXeX`) What are the benefits of specifying commands in the form of sound? Why not translate the sound to text first and proceed with the text?**

Given the success of current automatic speech recognition (ASR), CLIP, and large language models (LLM), we agree that these models could be used as modules of an alternative system. However, we would like to justify our choice to directly use sound commands from the following three aspects.

- Simplified fine-tuning. Our method does NOT assume that a sound-to-text module is perfect. In practice, the transcribed text from ASR and similar sound recognition modules is not always perfect and could be erroneous [22]. Fine-tuning ASR and subsequent intent understanding module requires expertise and ground-truth transcriptions, which is difficult for untrained non-expert users to provide. The third column in Table 1 and line 285-287 show the inefficiency of such a system during finetuning (ANR baseline). In contrast, by combining the sound processing unit into the representation, Dif-VAR allows end-to-end visually-guided updates of the sound recognition using the user’s own voice and knowledge without the need for transcriptions, as we detailed in line 189 - 193.

- Reduced intermediate errors and better performance. Our choice of skipping sound-to-text transcriptions is inspired by recent research in end-to-end spoken language understanding (E2E-SLU), as introduced in line 62 - 66. An E2E-SLU extracts user intent directly from speech without using the text. Such a system outperforms the ASR+NLU pipeline [17], coinciding with our results of the ANR baseline, as shown in Table 1 and Table 2 and analyzed in line 254-256.In this work, we intend to introduce this end-to-end intent understanding to the robotics community.

- Wider applications. In our work, voice commands are not limited to speech or language. Environmental sounds like alarm clock sounds and dog-barking sounds, musical tones, and emotions and background noise in a speech can all indicate a command but are ignored by a common sound-to-text module. In our Desk environment, the last row in Fig. 11 and Appendix D.2 shows that the robot is able to press the button on the desk to mute an alarm given an alarm clock sound. In Table 1, Row environment, NSynth dataset, we also show that when given musical tones or dog-barking sounds, the robot is able to reach the corresponding block.


**References:**

[17] M. Kim, G. Kim, S.-W. Lee, and J.-W. Ha. St-bert: Cross-modal language model pre-training for end-to-end spoken language understanding. In International Conference on Acoustics, Speech, and Signal Processing (ICASSP), pages 7478–7482, 2021.

[22] Y. Tada, Y. Hagiwara, H. Tanaka, and T. Taniguchi. Robust understanding of robot-directed speech commands using sequence to sequence with noise injection.  Frontiers in Robotics and AI, 6:144, 2020

---

### Decision · Program_Chairs · 2023-08-30

**Decision:**

Accept (Poster)

**Comment:**

The paper proposes a method for reward learning, which specifically gives a reward function that is (i) conditioned on a user audio command, and (ii) also a function of the current environment image.  In this sense, the reward function is a visual-audio one, where the audio specifies the task, and the image provides the current environment observation, which together determine the reward.  The reward learning method is combined with RL to learn policies, and tested on a few different simulated environments, and in a type of multi-object reaching task in the real world.

The reviewers appreciated various aspects about the paper, including the overall goal of having audio-specified goals, and that "reward design is a pain for RL and data efficiency is a huge problem." (S2KB). There were comments from multiple reviewers that perhaps just running state-of-the-art speech-to-text, and/or involving LLMs, could be suitable, but in the rebuttal the authors I believe have argued convincingly that using audio processing that doesn't go through text has several beneficial aspects, including being able to capture non-linguistic sounds, and potentially having less intermediate errors.  Some reviewers appreciated this more "end-to-end" way of doing audio-specified rewards.  Aside from performance implications, it is also interesting and merits study.

There are some weaknesses as well that were pointed to, some of which were addressed in rebuttal responses.  In general, although I personally can appreciate some of the challenges of the real-world example (uncalibrated camera, no known-object assumptions), the demo does come off as looking simplistic, in part because it seems essentially a 1-degree-of-freedom task (I know it's "2", but it's basically 1) with no contact.  Still, the reviewers thought the sim evaluations "drove a point" (b57y) and the real-world results were better than having no real-world results.  The work does seem incremental with respect to VAR, which is not necessarily a bad thing, and the authors clarified that there are a couple practical benefits of DifVAR including not needing triplet-paired data.

Overall, the authors engaged in good discussion with the reviewers.  One reviewer (TXeX) increased their score to Weak Accept after robust responses by the authors to their questions.  Another reviewer (Tifq) also increased their score to Weak Accept after discussion with reviewers and the AC after which they decided their concerns were orthogonal to the main point of the paper.  Another reviewer (Tifq) had multiple rounds of questions, and were satisfied by the authors' responses.

Also, it should be noted that most roboticists do not typically use audio signals, and especially not typically for task specification in a way that doesn't first get converted to text.  I think this is a plus for the paper, that it addresses a type of problem that is not that common.

Overall, I think the paper will be of sufficient net benefit to the robot learning community to recommend its acceptance.  Please include the updated plots on sample efficiency, and the comparison with the Whisper-based method, in the final camera ready version.